

# Prediction of junior faculty success in biomedical research: comparison of metrics and effects of mentoring programs

Christopher S. von Bartheld[1], Ramona Houmanfar[2] and Amber Candido[2]

[1] Department of Physiology and Cell Biology, University of Nevada School of Medicine, Reno, NV, USA
[2] Department of Psychology, University of Nevada - Reno, Reno, NV, USA

## ABSTRACT

Measuring and predicting the success of junior faculty is of considerable interest to faculty, academic institutions, funding agencies and faculty development and mentoring programs. Various metrics have been proposed to evaluate and predict research success and impact, such as the h-index, and modifications of this index, but they have not been evaluated and validated side-by-side in a rigorous empirical study. Our study provides a retrospective analysis of how well bibliographic metrics and formulas (numbers of total, first- and co-authored papers in the PubMed database, numbers of papers in high-impact journals) would have predicted the success of biomedical investigators ($n = 40$) affiliated with the University of Nevada, Reno, prior to, and after completion of significant mentoring and research support (through funded Centers of Biomedical Research Excellence, COBREs), or lack thereof (unfunded COBREs), in 2000–2014. The h-index and similar indices had little prognostic value. Publishing as mid- or even first author in only one high-impact journal was poorly correlated with future success. Remarkably, junior investigators with >6 first-author papers within 10 years were significantly ($p < 0.0001$) more likely (93%) to succeed than those with ≤6 first-author papers (4%), regardless of the journal's impact factor. The benefit of COBRE-support increased the success rate of junior faculty approximately 3-fold, from 15% to 47%. Our work defines a previously neglected set of metrics that predicted the success of junior faculty with high fidelity—thus defining the pool of faculty that will benefit the most from faculty development programs such as COBREs.

Corresponding author
Christopher S. von Bartheld,
cvonbartheld@medicine.nevada.edu

## INTRODUCTION

Faculty development has become the topic of considerable interest, with universities increasingly implementing formal mentoring programs to ensure that new faculty find suitable mentors and receive other help with career development (*Thorndyke et al., 2006*; *Bland et al., 2009*; *Steinert et al., 2009*; *Bruce et al., 2011*). In addition, federal agencies

(e.g., National Institutes of Health, NIH) are funding faculty development programs to reduce disparities in the geographic localization of biomedical research (*Process Evaluation of the COBRE Program, 2008*), to enhance diversity of the workforce (*Page, Castillo-Page & Wright, 2011*), and to promote clinical and translational research (*Meagher et al., 2011*; *Knapke et al., 2013*; *Knapke et al., 2015*). Sponsors such as the NIH have been collecting metrics on these programs (*Process Evaluation of the COBRE Program, 2008*; *Committee to Evaluate the EPSCoR, 2013*), showing success by various parameters. However, little attention has been directed either at the NIH or in evaluations of institution-based mentoring programs (*Bruce et al., 2011*; *Wadhwa, Das & Ratnapalan, 2014*) as to which metrics may predict success of the junior faculty member, and which junior faculty members will most likely benefit from these programs and succeed. Faculty can be divided in three groups: those who don't need additional support to succeed; those who will never obtain independent research funding, regardless of the effort and money invested in their mentoring; and the ones that fall between these two extremes. The latter are the ones who benefit the most from a mentoring program if it brings them across the threshold of where they can obtain independent status and maintain a productive, externally funded lab. But how does one identify these groups in advance—how does one predict future success?

## Metrics to predict success and impact

Multiple types of metrics have been claimed or proposed to measure academic research success and impact, including publication record: many publications, early publications, papers published in journals with high impact factors (IFs), first author papers, any co-authored papers, papers with high citation rates—as first author or as any (co-)author (*Khanna & Mouw, 1993*; *Symonds, 2004*; *Hirsch, 2007*; *Acuna, Allesina & Kording, 2012*; *Laurance et al., 2013*; *Carpenter, Cone & Sarli, 2014*; *Van Dijk, Manor & Carey, 2014*). One metric that has been gaining considerable popularity, due to its simplicity, is the h-index (*Hirsch, 2005*). This index is used to measure the research impact of scientists, and it is provided in major databases such as Web of Science and Scopus (*Bakkalbasi et al., 2006*; *Falagas et al., 2008*). However, a major criticism of the h-index has been that it does not distinguish whether an author is the first or last author (both of whom together should get most of the credit), or just one of potentially hundreds of co-authors with minimal contributions (*Schreiber, 2008*; *Bornmann & Daniel, 2009*; *Romanovsky, 2012*; *Biswal, 2013*; *Carpenter, Cone & Sarli, 2014*). This has led to multiple proposals of revised indices (*Bornmann et al., 2011*; *Biswal, 2013*; *Carpenter, Cone & Sarli, 2014*).

Since major obstacles to a meaningful program evaluation are "selection bias" and the lack of a suitable control group for comparison (*Sambunjak, Straus & Marušić, 2006*; *Bruce et al., 2011*; *Steinert, 2012*; *Committee to Evaluate the EPSCoR, 2013*), we took advantage of the fact that our institution has competed in the last 15 years in multiple rounds of applications for Centers of Biomedical Research Excellence (COBREs), with four completed successful COBREs and four unsuccessful applications (with different teams of junior faculty). This provided a sufficient "$n$" of faculty ($n = 40$) to assess in a

retrospective study the utility of metrics that have been proposed to predict faculty success and to quantify the impact of a substantial federally funded mentoring program (COBRE).

## MATERIALS & METHODS

### Faculty inclusion criteria

This project was reviewed by the University of Nevada, Reno Social Behavior and Education IRB, and it was determined that this work does not constitute human subject research and does not require human research protection oversight by the IRB. We compiled metrics and examined and compared the bibliographic output and grant support of all those junior faculty at the University of Nevada, Reno, that were proposed between 2000 and 2014 to become project leaders in a COBRE mentoring program and were actually supported for at least two years in such a program (="mentored group"), or were proposed for such a position but the COBRE was not funded (control group). We will refer to the COBRE-supported group as the "mentored group," but caution the reader to keep in mind that mentoring is only part of the benefits of COBRE support. COBREs provide up to five years of research funding for junior faculty (phase I), and each COBRE can be competitively renewed after five years for a second five-year period (phase II). Among our institution's ∼90 biomedical faculty, approximately 30–35 are eligible to become COBRE project leaders. Between 2000 and 2014, four COBREs were funded, two completed phases I and II, two are currently in phase I, and four proposed COBREs were not funded, providing a total of $n = 40$ faculty, with $n = 20$ for the mentored group, and $n = 20$ for the control group. When an unfunded COBRE was resubmitted in the following year with the same team of junior faculty, only the first year of submission was considered, to not duplicate results. Occasionally, junior faculty moved from a phase I COBRE to a phase II COBRE ($n = 5$; only two faculty completed a total of 10 years of COBRE funding), or moved from an unfunded COBRE to a funded COBRE ($n = 3$), or were included in different years on two different unfunded COBREs ($n = 2$), but those were a relatively small proportion of the total number of junior faculty ($n = 40$). Junior faculty were defined according to COBRE rules as faculty who had not yet served as principal investigator on an NIH R01 grant or a comparably large grant project.

### Metrics

Entire PubMed bibliographies were established for each of the faculty. Moreover, all citations as per Web of Science$^{TM}$ Core Collection, Basic Search, were compiled for the year up to and including the proposed start date of the COBRE to determine the h-index (*Hirsch, 2005*) at that time, i.e., the start of mentoring/research support (or proposed, if not funded). A scientist has index h if h of his/her $N_p$ papers have at least h citations each, and the other ($N_p$-h) papers have no more than h citations each (*Hirsch, 2005*). The same data were used to calculate another index, the Ab-index that takes into account multiple authorship (*Biswal, 2013*). In addition, the number of 1st-author papers during the 10-year period preceding the proposed COBRE start date (year) was determined for each faculty. In one case, a faculty member had already made the transition from 1st-author publishing

to last-author publishing; in this case, the number of last-authored papers was counted instead of the number of 1st-authored papers for the same 10-year period. We also counted the number of first-authored and the number of co-authored (middle author) papers in PubMed with a high (>9) journal impact factor. Finally, the number of last-author papers was determined for all faculty during and upon graduation from the COBRE program (or a comparable time frame when the COBRE was not funded). All investigators considered here followed the convention where senior authorship (corresponding authorship) is expressed by last-author position among authors. A successful faculty was defined as having external (not only COBRE-funding) of any amount and duration (in all of our cases at least two years of funding), and in addition publishing on average at least one last-author (=senior author) paper in PubMed per year during or upon graduation from the COBRE (or a comparable time frame when the COBRE was not funded). This definition combines two key features of how academic centers evaluate faculty's research success: extramural funding and peer-reviewed publications (*Kairouz et al., 2014*). Senior-author publications during COBRE support were not counted towards this number if other external grant support was lacking, in order to evaluate true independence from COBRE support. Note that this definition does not require that the successful faculty remain located in the USA (occasionally, project leaders were offered and they accepted academic positions outside the USA). Simply staying on COBRE funds for extended periods of time without other external grant support was not considered an independent, externally funded and successful investigator in biomedical research. NIH grant awards were verified in the public database, NIH Reporter (http://projectreporter.nih.gov/reporter.cfm), and information about other external grant funding was obtained by grant citations. Information about the ranking of the University of Nevada, Reno was obtained from www.arwu.org, and statistics software used was SPSS version 23 (IBM, Armonk, New York, USA).

## RESULTS

### Characterization of groups

We first compared the metrics that were compiled for the two groups, the control vs. mentored (=COBRE-supported) groups, to verify that they were essentially equivalent. As shown in Table 1, the h-indices for the two groups of faculty at the start of mentoring (or lack thereof) were very similar—10.65 vs. 12.00 (statistically not different) and the Ab-indices were 177.6 vs. 148.6 (again statistically not different). The mean number of total papers in PubMed during the 10 years prior to proposed mentoring start, per faculty, was 14.9 vs. 17.5, again not a significant difference. The number of 1st-author papers in PubMed differed by less than 2.5 (6.85 vs. 4.45), statistically insignificant (see Table 1). The number of first author papers in high-impact journals, defined as journals with an impact factor (IF) >9, was a mean of 0.9 for the control, and a mean of 0.8 per faculty in the mentored group. The number of co-authored papers in high impact journals was 2.35 for the control, and 1.35 per faculty in the mentored group (statistically insignificant). The gender of junior faculty was 50% female vs. 50% male for the control, and 25% female vs. 75% male for the mentored group. The delay between first publication (in PubMed)

**Table 1** Characteristics of the two groups of junior faculty (Control = no COBRE; Mentored = with COBRE support) at the University of Nevada, Reno, 2000–2014.

| | n | h-index | Ab-index | # of all papers | # of 1st author papers | # of 1st author papers with high JIF | # of co-author papers with high JIF | M/F ratio | Ethnicity: Caucasian/ Asian[*] | English as 1st language |
|---|---|---|---|---|---|---|---|---|---|---|
| Control: no COBRE | 20 | $10.65 \pm 1.27$ | $177.6 \pm 37.8$ | $14.9 \pm 2.3$ | $4.45 \pm 0.89$ | $0.90 \pm 0.33$ | $2.35 \pm 0.32$ | 10/10 | 18/2 | 10/20 |
| Mentored: with COBRE | 20 | $12.00 \pm 1.23$ | $148.6 \pm 27.0$ | $17.5 \pm 2.0$ | $6.85 \pm 0.95$ | $0.80 \pm 0.27$ | $1.35 \pm 0.61$ | 15/5 | 14/6 | 12/20 |
| Statistics (*t*-test) | | $p = 0.449$ | $p = 0.610$ | $p = 0.394$ | $p = 0.073$ | $p = 0.816$ | $p = 0.156$ | | | |

**Notes.**

Values $\pm$ standard error of the mean (SEM); *p*-values are for unpaired *t*-test.

COBRE, Center of Biomedical Research Excellence; JIF, Journal impact factor; M/F, male/female.

[*] Only two ethnic categories were involved.

**Table 2 Comparison of junior faculty success rates and outcomes (Control = no COBRE; Mentored = with COBRE support) at the University of Nevada, Reno, 2000–2014.**

| | Overall faculty success | Male faculty success | Female faculty success | English as 1st language success | English as 2nd or 3rd language success | Retention at UNR | Successful faculty: mean # of 1st author papers | Faculty without success: mean # of 1st author papers |
|---|---|---|---|---|---|---|---|---|
| No COBRE | 15% 3/20 | 30.0% 3/10 | 0.0% 0/10 | 10% 1/10 | 20% 2/10 | 40% 8/20 | 11.33 $n = 3$ | 3.24 $n = 17$ $p = 0.19$[**] |
| With COBRE | 47.1% 8/17 | 66.6% 10/15 | 20.0% 1/5 | 45.5% 5/11 | 75% 6/8 | 64.7% 11/17 | 13.00 $n = 11$ | 3.33 $n = 9$ $p = 0.0002$[**] |
| Total $n$ | 37[*] | 25[*] | 15[*] | 22 | 18 | 37[*] | 14 | 26 |
| Statistics ($t$-test) | $p = 0.0404$ | | | | | $p < 0.1$ | | $p < 0.0001$ |

**Notes.**

[*] $n = 37$ (not 40) for the *overall* success and retention calculation, because three successful faculty from the two current COBREs cannot yet be compared with their peers who are still being mentored.

[**] unpaired $t$-test; when all successful vs. non-successful faculty combined for # of 1st-author papers, $p < 0.0001$.

COBRE, Center of Biomedical Research Excellence; UNR, University of Nevada, Reno.

and proposed COBRE start date was 14.0 years for the control, and 10.9 years for the mentored group (statistically insignificant difference, Table 1). Thus, the two groups were well-matched by h-index and Ab-index, with the investigators on funded COBREs having published slightly more 1st-author papers, but having published less of these 1st-author papers (and co-authored papers) in high impact factor journals than the investigators proposed for unfunded COBREs.

## Success rate of junior faculty

The success rate, defined as external funding and at least one paper in the PubMed database per year (on average) as senior author, was $3/20 = 15\%$ for faculty in unfunded COBREs (the "baseline", $n = 20$) and $8/17 = 47.1\%$ ($n = 17$) after COBRE-support and mentoring (see Table 2, a significant difference with $p = 0.0404$). For the purposes of calculating the success rate, we did not include faculty of the *current* COBREs at the University of Nevada, Reno, (since it is too early to evaluate them—for this reason there is an $n = 17$ for the comparison of success rates). Accordingly, the COBRE support and mentoring increased the success of faculty by over 3-fold (or by 213%). Male faculty were more successful than female faculty. Faculty with English as 2nd language had more success ($8/18 = 44.4\%$) than native English speakers ($6/22 = 27.3\%$). The fraction of ethnic minorities was too small for any meaningful conclusions. Previous studies have indicated that mentoring programs increase the retention of faculty (*Wingard, Garman & Reznik, 2004*; *Sambunjak, Straus & Marušić, 2006*; *Bland et al., 2009*). Among the mentored faculty (1–15 years after onset) retention was 64.7% (11/17), up from a baseline of 40.0% (8/20) (Table 2, although this increase was not statistically significant, $p > 0.1$). The difference in delay between first publication and proposed COBRE start date (14.0 years for the mentored group; 10.9 years

for the control group) cannot explain the difference in success, because the successful faculty had a mean delay of 10.8 years, while the non-successful faculty had a mean delay of 13.4 years, so the non-successful faculty actually had more years of experience than the successful faculty. COBRE mentoring reduced the time required to obtain external funding from 3.9 to 3.4 years, but the decrease was not statistically significant. For successful faculty (per our definition of success), COBRE mentoring reduced the time between proposed start of mentoring and external funding from a mean of 4.0 to 3.0 years (not statistically significant).

### Utility of metrics for prediction of success

We first tested the h-index, since it is widely believed to have predictive power (*Hirsch, 2007*; *Acuna, Allesina & Kording, 2012*; *Van Dijk, Manor & Carey, 2014*). The h-index at the start of proposed COBRE funding was $12.00 \pm 1.28$ (SEM) for successful faculty, and $10.65 \pm 1.27$ for non-successful faculty (insignificant difference); accordingly, the h-index did not distinguish between successful and unsuccessful faculty. When we plotted the h-indices within sub-groups, there was no predictive value, except for a trend for more success with mid-level h-indices (see Fig. 1). The Ab-index, which proportions citations according to the author rank (*Biswal, 2013*), was $148.6 \pm 28.2$ for the mentored group, and $177.6 \pm 37.8$ for the control group, with no statistically significant difference. Accordingly, even when controlled for citations as co-author, the index did not distinguish between successful and non-successful faculty. When the Ab-index was plotted according to sub-groups, a bi-phasic curve emerged, with the lowest chance of success for the lowest and highest Ab-indices (0–50 and above 500), and the best chance of success (nearly 60%) in the range of 75–200 (see Fig. 1).

### Examination of authorship metrics

The total number of papers published in PubMed (including co-author, 1st-author, and last-author papers at the start of COBRE funding or the *proposed* start date) showed a trend from generally lower success rates with low numbers of papers (0–15), and higher success rates with larger numbers of papers (15–45, Fig. 2), but the pattern did not reflect and discriminate between the successful and the non-successful faculty, and therefore would be of limited use to identify groups. Faculty success has been linked to publishing in top journals (*Symonds, 2004*; *Carpenter, Cone & Sarli, 2014*; *Van Dijk, Manor & Carey, 2014*). We therefore examined the numbers of papers in high-impact factor (IF > 9) journals. There was no difference between chances of faculty for success with none or one 1st-author high impact journal paper, but when the number of 1st-author papers in journals with high impact factors increased to two or more, there was a strong correlation with success ($r^2 = 0.703$, Fig. 2). The number of co-authored (middle author) papers in high impact journals did not correlate with faculty's success—in fact, successful faculty published fewer high impact papers as co-authors (1.50 per faculty) than non-successful faculty (2.04 per faculty, statistically insignificant). Accordingly, publishing as a co-author in high impact journals does not increase chances of success, but publishing multiple 1st-author papers in high impact journals does.

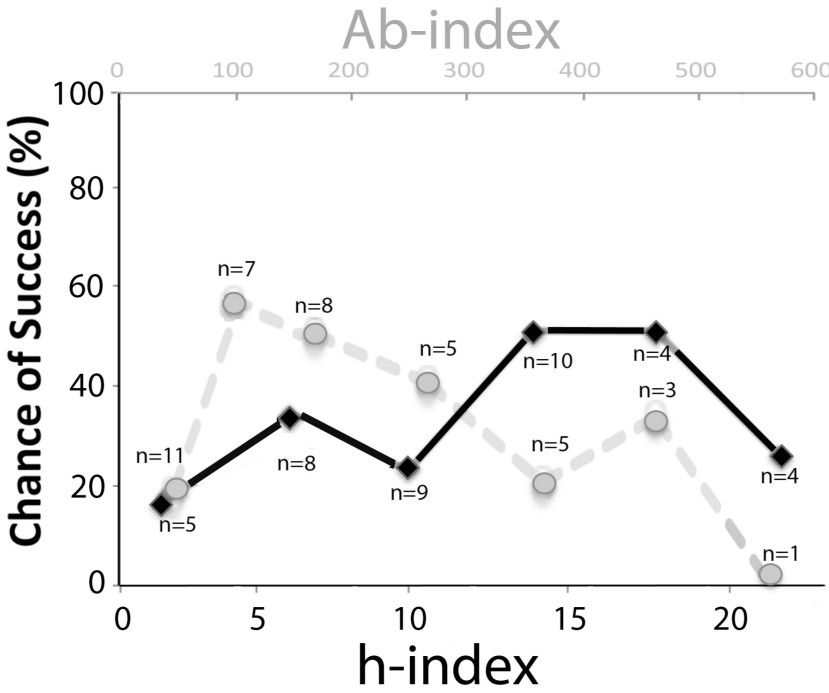

**Figure 1 Chances of junior faculty success plotted as a function of the h-index and the Ab index.** The chances of success of junior faculty at the University of Nevada, Reno from 2000 to 2014 are plotted as a function of the h-index (*Hirsch, 2005*) (in increments of 3–5, black font, lower *x*-axis) at the time of proposed start of COBRE mentoring. In addition, the chances of success are plotted as a function of the Ab-index (*Biswal, 2013*) that takes co-authorship into account (in increments of 50–100, grey, upper *x*-axis). The number of determinations for each data point is indicated (*n*). Neither of these indices predicted success with more than 60%, even for selected subgroups, and dropped off at higher values. Total $n = 40$, with $n = 14$ successful and $n = 26$ not successful.

Next, we examined whether the number of 1st-author papers in any PubMed-listed journal (regardless of the journal's impact factor) would predict success. There was a near perfect ($38/40 = 95.0\%$ correct) prediction of success, showing a very small chance ($1/24 = 4.1\%$) of success for faculty with six or less 1st-author papers in the 10 years prior to the year of proposed COBRE funding, and a very high chance ($13/14 = 92.9\%$) of success for those with seven or more 1st-author papers (see Fig. 3). Accordingly, the number of 1st-author papers in PubMed distinguishes with high precision ($p < 0.0005$) between successful and non-successful junior faculty. We conclude that, among all examined metrics, the number of 1st-author papers in the preceding 10 years is the most powerful predictor of biomedical research success, as per our definition of faculty success.

## DISCUSSION

Our study is the first retrospective, empirical study that compares the long-term effectiveness of a faculty mentoring and research support program with a suitable control group. Utilizing an appropriate control group is required, but often lacking in the research design of program evaluations (*Jacobi, 1991*; *Morzinski & Fisher, 1996*; *Steinert, 2000*; *Sambunjak, Straus & Marusić, 2006*; *Steinert, 2012*). Furthermore, we identify a simple,

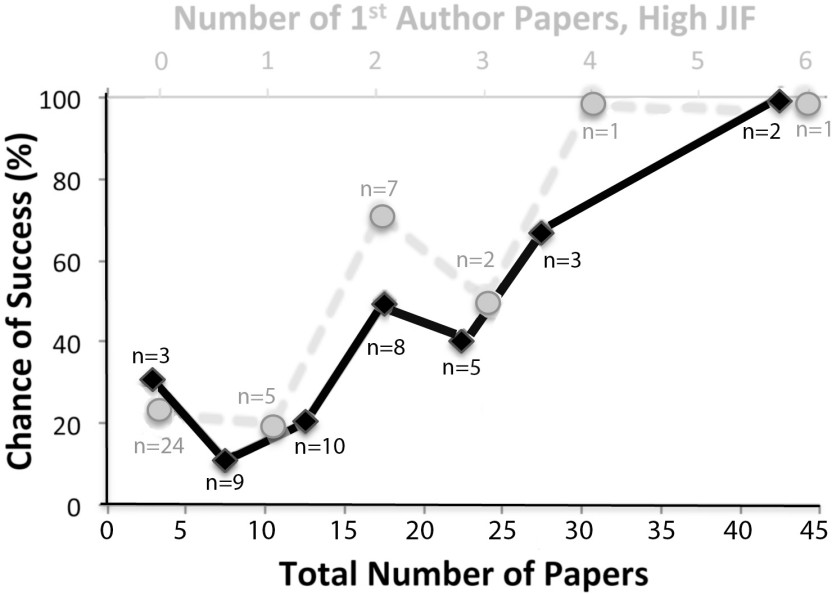

**Figure 2 Chances of junior faculty success plotted as a function of previously authored PubMed papers.** The chances of success of junior faculty at the University of Nevada, Reno from 2000 to 2014 are plotted as a function of the number of authored or co-authored papers in PubMed and published prior to the time of proposed start of COBRE mentoring (in increments of 5, black font, lower $x$-axis). In addition, the chances of success are plotted as a function of the number of 1st-author papers in journals with high impact factors (IF > 9) (in increments of 1, grey, upper $x$-axis). The number of determinations for each data point is indicated ($n$). Chances of success increased with larger numbers, but they reached 100% only with the highest values, limiting a meaningful prediction to a small percentage of faculty. Total $n = 40$, with $n = 14$ successful and $n = 26$ not successful.

easy-to-use metric that predicts faculty success much more reliably than several previously proposed indices, such as the h-index and the Ab-index. Our work helps to define a pool of junior faculty that presents a prime target for faculty development resources, and, once externally validated, should make such programs more efficient and successful.

## Measuring effectiveness of faculty development programs

Previous attempts to define mentoring success have been criticized for lack of adequate control groups and for relying on self-report and participant satisfaction—essentially testimonials and opinions (*Sheets & Schwenk, 1990*; *Jacobi, 1991*; *Morzinski & Fisher, 1996*; *Buddeberg-Fischer & Herta, 2006*; *Sambunjak, Straus & Marušić, 2006*; *Thorndyke, Gusic & Milner, 2008*; *Bruce et al., 2011*). One major novel feature of our study is the comparison with a near optimal control group—junior faculty who were deemed sufficiently competitive to be included in a COBRE grant proposal as project leaders. This is about as close as ethically possible to a truly randomized trial. Accordingly, our two groups were not likely to be skewed by "selection bias" (*Jacobi, 1991*; *Bruce et al., 2011*).

Our study validates the design and implementation of NIH-funded mentoring and research support, especially the COBRE faculty development program, which we show increases faculty success by more than 3-fold. This amount of increase is somewhat larger than that (2.4-fold increase) reported for another mentoring program, the NIH-funded

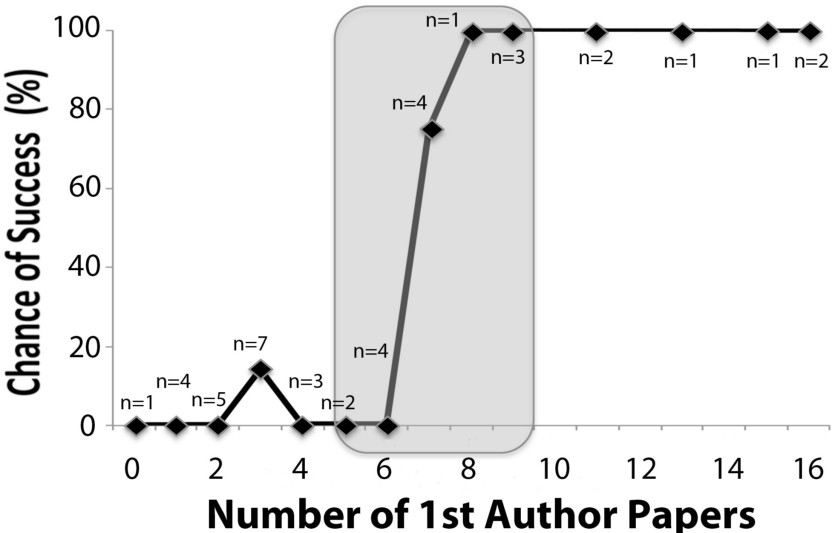

**Figure 3 Chances of junior faculty success as a function of previous 1st-authored papers in PubMed.** The chances of success of junior faculty at the University of Nevada, Reno from 2000 to 2014 are plotted as a function of the number of 1st-author papers listed in PubMed and published during the 10-year period prior to the date of the proposed start of COBRE mentoring (in increments of 1, $x$-axis). The success rate flips from 1/24 (4.1%) to 13/14 (92.9%) between six and seven 1st-authored papers in PubMed. Each data point represents between 1 and 7 faculty, total $n = 40$. The number of 1st-authored papers in the preceding decade predicted outcome in 38/40 (95%), and the difference of mean numbers of such papers per faculty, between successful and not successful faculty, was significant with $p < 0.0001$ (successful $n = 14$, not successful $n = 26$). The shaded area from five–nine 1st-authored papers defines the pool of junior faculty to benefit the most from a COBRE.

program by the Advanced Research Institute (ARI) in geriatric mental health at Cornell University (*Bruce et al., 2011*). This two-year program targets scholars nationwide that are midway through a mentored career development award. The authors of the ARI program, however, did not compare their results with a local control group within a similar academic environment. To our knowledge, an average increase in junior faculty success of over 3-fold is the first "hard" number obtained in a controlled study for a faculty development program (COBRE) that invests $170 million annually and over $2 billion since inception in 2000 (*Committee to Evaluate the EPSCoR, 2013*). Accordingly, our study lends support to the effectiveness of this NIH-funded program.

Female junior faculty were less successful (as defined above) than male faculty at our institution (see Table 2)—similar to previous studies (*Laurance et al., 2013*; *Van Dijk, Manor & Carey, 2014*). Time constraints during a critical period of a scientist (grad student/post-doc productivity) may compete with child-bearing and responsibilities of raising a family (*Ceci & Williams, 2011*), but this is a complex issue and additional factors, including gender bias in the academic community, have been well documented to be relevant factors (*Fried et al., 1996*; *Steinpreis, Anders & Ritzke, 1999*; *Moss-Racusin et al., 2012*). When female faculty published beyond the critical 6 first-authored papers, they appeared to be on track for success, comparable to their male counterparts, suggesting that the obstacles are at earlier rather than later stages of career development.

## Prediction of junior faculty success

Attempts to predict faculty success or any trainee's success have a long history (*Creswell, 1986*; *Jacobi, 1991*; *Khanna & Mouw, 1993*; *Olds, 2001*; *Symonds, 2004*; *Hirsch, 2007*; *Acuna, Allesina & Kording, 2012*; *McNutt, 2014*; *Van Dijk, Manor & Carey, 2014*; *Wadhwa, Das & Ratnapalan, 2014*). Recent work has proposed to utilize various metrics to define who is most likely to succeed. These metrics have included aspects of the publication record: number of publications, early publications, papers published in journals with high impact factors (IFs), papers published in *Science* or *Nature* (*Symonds, 2004*), 1st-author papers, any co-authored papers, papers with high citation rates—as first author or as any (co)author (*Acuna, Allesina & Kording, 2012*; *Laurance et al., 2013*; *Van Dijk, Manor & Carey, 2014*), but also aspects of training (reputation of the advisor, lab, or university for graduate work or post-graduate work (*Acuna, Allesina & Kording, 2012*; *Laurance et al., 2013*; *Van Dijk, Manor & Carey, 2014*), as well as factors such as gender, with being male giving a significant advantage (*Laurance et al., 2013*; *Van Dijk, Manor & Carey, 2014*).

Being a native English speaker is also thought to convey a small, but measurable boost, albeit with great variation among individuals (*Laurance et al., 2013*). This seems to be a discrepancy with our finding of second language English speakers being more successful than native English speakers. We considered that two differences between studies may be relevant: Laurance and colleagues studied academicians throughout the world (where many languages compete), while our study was at one university in the USA (where second language English speakers compete primarily with native English speakers). Furthermore, Laurance and colleagues measured total number of publications (regardless of first-author status), while we report the success of those faculty with a large number of 1st author publications. To determine whether the latter difference plays a role, we compared the variable of native English vs. English as second language for all publications in our database. Our analysis indicates that faculty with English as second language published more papers (counting any type of authorship) than native English speakers, in both the COBRE-supported and control groups: 18.0–20.75 per investigator vs. 12.3–15.25 papers per investigator.

Taking various metrics into account, predicting h-index into the future is only 0.48–0.67 effective (*Acuna, Allesina & Kording, 2012*), or 0.62–0.74 (for the single most predictive metric = 0.62, and 37 features combined = 0.74) (*Van Dijk, Manor & Carey, 2014*), while our single metric of 1st-author papers within the preceding 10 years yields 0.95 predictive value of success. Based on our formula, we would have predicted that only 2/20 proposed junior faculty in unfunded COBREs would ultimately succeed (3/20 actually did), while we would have predicted that 12/20 proposed junior faculty on funded COBREs would succeed (when 11/20 actually succeeded)—scoring a 38/40 correct. Therefore, our study corroborates the evaluation and funding decisions by NIH for our institution's COBRE grant submissions: only COBREs with the more promising junior faculty were funded.

There are anecdotal reports that the potential of a trainee to become a successful independent investigator can be judged early in the training experience (*Creswell, 1986*; *Olds, 2001*). However, to our knowledge, how and whether such a prediction can be

substantiated by quantifiable parameters (metrics) has not been explored, although metrics have recently been employed to calculate chances of securing a job in academia (*Acuna, Allesina & Kording, 2012*) or to become a principal investigator based on authorship in databases (*Van Dijk, Manor & Carey, 2014*). Interestingly, our analysis supports the notion that several 1st-author papers can make up for lack of publications in top journals (*Van Dijk, Manor & Carey, 2014*).

Another interesting finding of our study is that having published in one high-impact journal, even as first author, does not confer a significant increase in future success, and publishing such papers as middle author does not increase success at all (as discussed above). Only two or more 1st-author papers in high impact journals increased the chance of success (see Fig. 2). This is contrary to common opinions and reports that tend to give candidates preference if they have one publication in a top journal (*Symonds, 2004*; *Acuna, Allesina & Kording, 2012*; *Schmid, 2013*; *Van Dijk, Manor & Carey, 2014*). Our study indicates that the increased intellectual involvement that 1st-authorship usually entails, provides the best guarantee of future success if it can be sustained over several years, even if these papers are not published in high impact journals. Accordingly, counting 1st-author papers may define a more dynamic pool of junior faculty consisting of the ones with the intellectual drive, mentality and persistence to become successful principal investigators (*Creswell, 1986*), while eliminating those who contribute but overall have less intellectual involvement. This is in agreement with a recent large-data study predicting principal investigator status (*Van Dijk, Manor & Carey, 2014*).

Limitations of our study are relatively small group sizes. We only assessed which metrics predicted success in our environment (mid-sized university in a small state). We cannot extrapolate our data for larger states, and elite universities. But our conclusions likely apply for most medical schools—the "99%" as opposed to the "1%" institutions, and therefore should be relevant for the large majority of academic medical centers. Future work will further probe the validity of the 10-year timeframe, refine the metrics during the transition from 1st-author to last-author publishing, and examine the impact associated with temporary leaving academic research, such as during maternity, teaching, or private sector employment.

## CONCLUSIONS

We show that a relatively simple metric—the number of 1st-author publications—far outperforms other metrics such as the h-index, journal impact factors, and citation rates in predicting research success of junior faculty. However, proxies alone are insufficient in evaluating or predicting faculty success, and further work is needed to determine which aspects of the COBRE and other faculty development programs contribute to success. Nevertheless, our study can now be replicated and validated at other biomedical institutions to predict the most suitable targets for faculty development and to evaluate and improve various types of mentoring programs.

# ACKNOWLEDGEMENT

The authors thank Dr. Treg Gardner for computational support, Andrea Agarwal for data management, and Drs. James Kenyon, Thomas Kozel, and David Westfall for helpful discussions.

### Funding

This study was supported by the National Institute of General Medical Sciences of the National Institutes of Health under grant number P20 GM103554. The content is solely the responsibility of the authors and does not necessarily represent the official views of the National Institutes of Health. The funders had no role in study design, data collection and analysis, decision to publish, or preparation of the manuscript.

### Grant Disclosures

The following grant information was disclosed by the authors:
National Institute of General Medical Sciences of the National Institutes of Health: P20 GM103554.

### Competing Interests

The authors declare there are no competing interests.

### Author Contributions

- Christopher S. von Bartheld conceived and designed the experiments, analyzed the data, wrote the paper, prepared figures and/or tables, reviewed drafts of the paper, had the institution's IRB determine that this work does not require human research protection oversight by an IRB.
- Ramona Houmanfar analyzed the data, reviewed drafts of the paper, had the institution's IRB determine that this work does not require human research protection oversight by an IRB.
- Amber Candido analyzed the data, prepared figures and/or tables, reviewed drafts of the paper.

### Human Ethics

The following information was supplied relating to ethical approvals (i.e., approving body and any reference numbers):

1. University of Nevada, Reno Social Behavior and Education IRB
2. This project was reviewed by the above-named IRB, and it was determined on November 25, 2014 [Reference #685509-1] that this program evaluation did not constitute human subject research and did not require human research protection oversight by the IRB.

### Data Availability

The authors cannot publish or deposit raw data of their study that would allow to identify individual investigators. The determination of the IRB that this study does not require human research protection by an IRB was made with the understanding that no data from individual researchers will be published, except in aggregates.

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
