# Peer review of "Prediction of junior faculty success in biomedical research: comparison of metrics and effects of mentoring programs"

_PeerJ, doi:10.7717/peerj.1262_

## Round 0.1 · original submission · Minor Revisions

The authors describe the study of the success of junior faculty enrolled in the National Institutes of Health-sponsored professional development program (COBRE) at the University of Nevada at Reno. They are the first to use this cohort in a study, and their work provides the tools for others to do studies looking at program success by evaluating faculty success (extramural funding, publication record). Especially important is that they compared the COBRE enrollees with non-COBRE enrollees at the same institution, a legitimate and very important “control” group usually lacking from evaluations of programmatic successes.

The two reviewers themselves work in the area of faculty development and program evaluation, and each suggests only minor revisions. One reviewer finds a context lacking as well as a superficial overview of issues impacting women faculty; there is a large literature on the latter that goes FAR beyond family issues. This reviewer would like some of the richer, deeper studies to be cited in the text and in the references. This reviewer also points out the discrepancy in findings based on English as a 1st vs. 2nd language; while this is an area for future research, the authors should emphasize this discrepancy more profoundly.

The other reviewer would have more information about the overall demographics of the “control” group of junior faculty in order to better validate the comparisons. This reviewer also applauds the description of the tools used for the evaluation as the authors make similar studies more readily reproducible based on their Methods.

This editor would add that the authors do a good job of describing the caveats in their conclusion that first author papers are the primary determinant of junior faculty success. I would suggest that in addition they make a strong warning that proxies are insufficient in evaluating faculty success (as their own work shows re: the h-index, for example), and that further work is needed to determine the range of COBRE aspects that contribute to this success.

·

Basic reporting

Article meets all expectations as a well organized and structured whole.

Experimental design

: The design of the original study is the first of its kind using COBREs funded and unfunded to define a measurement population, using those unfunded as a control group. Having a control group is exceptional. The structure is tight and self-reinforcing, the authors present their population as free from “selection bias” by using this model. Previous measures of success such as the h-index and the Ab-index were dismissed in this population as statistically insignificant with no predictive value.

Validity of the findings

This study validates the design and implementation of the NIH COBRE Program, in their medical center, it has increased faculty success by 3-fold plus. Since the results showed a higher success rate than the ARI program studied at Cornell, it would have been helpful to actually refer to the ARI study. The significant issue is their conclusion that the number of 1st-author publications far outperforms other metrics is true for this defined population. What is missing is the context: How many faculty out of the total were involved in COBRE applications? Were there outliers not involved in COBRE? What is it about the successful COBREs which furthers publishing productivity? Future work seems to be completely related to further validating their present study, not examining or discussing context, other possible effects such as prestige of graduate/postgraduate school where students are expected to publish at a high rate and have the experimental facilities to do so. If this is true then a “pre-selection bias” can exist among the 1st-author publications of the high performing group. Other variables could be included such as 1st language.

Additional comments

Measured outcomes are themselves variables: male faculty are more successful than female faculty p.8-9. On p. 13 the entire issue of gender is dismissed with one citation attributing time constraints to childbearing and family raising. This is an appalling lack of knowledge about female success in academia—implicit bias, discrimination, stereotyping all play a role. It reflects a common bias also—women have children and are therefore less productive, less worthy of putting resources into. There is an oddity not discussed at all: P.9 in your study you report faculty with English a 2nd language had more success than native English speakers in your population. Yet on p.14 when discussing the work of others you mention that being a native English speaker conveyed a small, but measurable boost, citing Laurance et al. 2013. Surely this is worthy of comment and possibly further study? Overall I enjoyed reading this article and think it significant, but with some reservations as stated above.

·

Basic reporting

Overall, I found the Bartheld and Candido article to be structured appropriately, clear and well-written. The authors provide a brief review of the literature on popular metrics (e.g. h-index and Ab-index) often used to predict scientific research impact or faculty success (pp.4-5) and challenges in measuring the effectiveness of faculty development and mentoring efforts. Furthermore, authors provide adequate details on methods so that the study could be replicated at other institutions with NIH Centers of Biomedical Research Excellence (COBRE) faculty development programs. However, authors could provide more detail on the characteristics of the comparison groups in Table 1 (e.g. race/ethnicity, in addition to gender and ESL). Additionally, the authors could improve article by clarifying the difference between “faculty development” and “mentoring” (p. 9)—though COBRE faculty development incorporates mentoring, these two terms shouldn’t be used interchangeably.

Experimental design

Bartheld and Candido offer an empirical retrospective study, over ten years in the making, comparing long-term effectiveness of a faculty mentoring (and research support) with comparable control groups. Authors make an excellent case for utilizing NIH COBRE faculty application teams at the University of Nevada at Reno, since they could obtain and analyze the bibliographic output and grant support for all participants (p.5). An important contribution of the design of this study is that it’s longitudinal and provides robust, reflective analysis of the impact of prior “faculty development” and/or “mentoring” on faculty success.

Also, authors note that one of the challenges to meaningful program evaluation has been the lack of suitable control groups for comparison. By comparing the COBRE faculty applications teams at Nevada (four successful and four unsuccessful teams), Bartheld and Candido are able to avoid selection bias and dissimilarities among participants. Thus, the value of this this study is strengthened by having comparable teams for comparison.

Again, I would prod the authors to provide additional details on the characteristics of the comparison groups in Table 1 (e.g. race/ethnicity, in addition to gender and ESL).

Still, I firmly believe this study as presented can be replicated by others.

Validity of the findings

The authors findings are statistically sound and impressive, concluding that the number of 1st author publications (regardless of journal's impact factor) can predict future faculty success. The authors admit the limitations of the study—the group sizes were extremely small, findings may not be generalizable to larger states or elite institutions, etc. (p.15), this study documents the benefits of COBRE for a certain set of early career faculty. Ultimately, I found this study to offer conclusive evidence for 1st authorship as a reliable predictor and the value of COBRE faculty development for faculty success over time.

Additional comments

Recommend this article be accepted with minor revisions.

---

## Round 0.2 · accepted · Accept

The authors have adequately responded to the reviewers' minor concerns.